**Data Availability Statement:** All relevant data are within the paper and its Supporting information files.

# Vitamin D related genetic polymorphisms affect serological response to high-dose vitamin D supplementation in multiple sclerosis

**Max Mimpen**[1], **Linda Rolf**[1], **Geert Poelmans**[2], **Jody van den Ouweland**[3], **Raymond Hupperts**[1,4], **Jan Damoiseaux**[5], **Joost Smolders**[6,7] *

**1** School for Mental Health and Neuroscience, Maastricht University, Maastricht, Netherlands, **2** Department of Human Genetics, Radboud University Medical Center, Nijmegen, Netherlands, **3** Department of Clinical Chemistry, Canisius-Wilhelmina Hospital, Nijmegen, Netherlands, **4** Department of Neurology, Zuyderland Medical Center, Sittard-Geleen, Netherlands, **5** Central Diagnostic Laboratory, Maastricht University Medical Center, Maastricht, Netherlands, **6** MS Center ErasMS, Departments of Neurology and Immunology, Erasmus University Medical Center, Rotterdam, Netherlands, **7** Department of Neuroimmunology, Netherlands Institute for Neuroscience, Amsterdam, Netherlands

* j.j.f.m.smolders@erasmusmc.nl

## Abstract

### Introduction

A poor 25-hydroxyvitamin D (25(OH)D) status is a much replicated risk factor for developing multiple sclerosis (MS), and several vitamin D-associated single nucleotide polymorphisms (SNPs) have been associated with a higher risk of MS. However, studies on the benefit of vitamin D supplementation in MS show inconclusive results. Here, we explore whether vitamin D-associated SNPs and MS risk alleles confound serological response to vitamin D supplementation.

### Methods

34 participants from the SOLARIUM study consented to genotyping, of which 26 had vitamin D data available. The SOLARIUM study randomised relapsing-remitting MS patients to placebo or 14,000 IU vitamin $D_3$ for 48 weeks. Participants were categorised as either 'carriers' or 'non-carriers' of the risk allele for 4 SNPs: two related to D binding protein (DBP) and associated with lower 25(OH)D levels (rs4588 and rs7041), and two related to vitamin D metabolism enzymes CYP27B1 and CYP24A1 and associated with a higher risk of MS (rs12368653; rs2248359, respectively). 25(OH)D levels were determined at baseline and after 48 weeks.

### Results

The DBP-related SNPs showed no difference in 25(OH)D status at baseline, but carriers of the rs7041 risk allele showed lower 25(OH)D-levels compared to non-carriers after 48 weeks of supplementation (median 224.2 vs. 332.0 nmol/L, p = 0.013). For CYP related

**Funding:** This study was funded by Nationaal MS Fonds grant OZ2016-001 and an unrestricted grant by Merck. The funders had no role in study design, data collection and analysis, decision to publish, or preparation of the manuscript.

**Competing interests:** MM has nothing to disclose; LR has nothing to disclose; GP is director of Drug Target ID, Ltd.; JO has nothing to disclose; RH received institutional research grants and fees for lectures and advisory boards from Biogen, Merck, and Genzyme-Sanofi; JD has nothing to disclose; JS received lecture and/or consultancy fees from Biogen, Merck, Sanofi-Genzyme, and Novartis. This does not alter our adherence to PLOS ONE policies on sharing data and materials.

SNPs, neither showed a difference at baseline, but carriers of the rs12368653 risk allele showed higher 25(OH)D-levels compared to non-carriers after 48 weeks of supplementation (median 304.1 vs. 152.0 nmol/L, p = 0.014).

## Discussion

Vitamin D-related SNPs affect the serological response to high-dose vitamin D supplementation. The effects on more common doses of vitamin D, as well as the clinical consequence of this altered response, need to be investigated further.

## 1. Introduction

Although the exact aetiology of multiple sclerosis (MS) remains unknown, many factors, both genetic and environmental, have been identified as contributors to the disease. Low levels of circulating 25-hydroxyvitamin D (25(OH)D) are a much replicated risk factor for both the pathogenesis [1–3] and disease course of MS [4–6]. Genome wide association studies show that some MS risk alleles are situated in or near genes coding for enzymes related to vitamin D metabolism [7]. Indeed, MS patients tend to have a genetic background that is associated with lower 25(OH)D levels [8], and low 25(OH)D levels are in turn associated with a higher risk of MS disease activity. [4–6]. Many studies have investigated the supplementation of vitamin D in order to correct this lower level of 25(OH)D. However, these studies show mixed results, with some showing no clinical benefit [9, 10], while others report a protective effect of high-dose vitamin D supplementation [11–14].

A study by Bhargava et al. showed a attenuated elevation of circulating 25(OH)D levels after vitamin D supplementation in MS patients, compared to healthy controls [15]. Recently, Graves et al. reported that vitamin D related single nucleotide polymorphisms (SNPs) associated with lower 25(OH)D levels, associate with an increased relapse risk in paediatric MS patients [16]. However, it remains unclear whether these genetic variations are confounding factors in clinical supplementation studies. In this respect, elucidating the effects of vitamin D-related genetic background on supplementation studies is important for the interpretation of the results from these studies. Therefore, we have evaluated the influence of several vitamin D metabolism-related polymorphisms on the serological response to high dose vitamin $D_3$ supplementation, in a cohort of interferon-beta treated relapsing-remitting MS patients from the SOLARIUM study [17].

## 2. Methods and materials

### 2.1 Patients

This study is a post-hoc extended analysis of the SOLARIUM study, which was a sub-study of the SOLAR study. The SOLAR study evaluated disease activity in interferon beta-treated RRMS patients using high dose vitamin $D_3$ supplements compared to placebo. The SOLARIUM study investigated the effect of high dose vitamin $D_3$ supplementations on the immune system composition. After randomisation, participants received either interferon-beta and a placebo or interferon-beta and vitamin $D_3$ supplements (cholecalciferol, Vigantol$^{\circledR}$Oil, Merck KGaA, Darmstadt, Germany) of 7,000 IU daily for 4 weeks, followed by 14,000 IU daily up to week 48. In- and exclusion criteria for the SOLAR and SOLARIUM studies are described elsewhere [13, 17]. Originally, genetic markers regarding vitamin D status were planned to be

gathered of all participants in the SOLAR study [18]. Ultimately, however, this analysis was not performed. Therefore, we acquired ethical approval to approach Dutch participants of SOLAR to acquire written informed consent for genetic analysis (Ethical Committee METC-Z, 11-T-03; Heerlen, the Netherlands). This resulted in 34 of 53 participants of the SOLARIUM study to provide consent for genetic analysis.

## 2.2 Vitamin D measurements

Plasma samples were collected at baseline and week 48 and stored at −80˚C. Plasma levels of 25(OH)D were measured using liquid chromatography-tandem mass spectrometry (LC-MS/MS) following earlier published procedures [19, 20]. Coefficients of variation (CV) were 7.4% at 36 nmol/L, 4.0% at 88 nmol/L, and 3.1% at 124 nmol/L, respectively. Lower limit of quantification was 1 nmol/L [21]. From the SOLAR dataset, serum 25(OH)D and 1,25(OH)D levels were available as determined using the DiaSorin immunoassay method, with an upper limit of 250 nmol/L [13].

Of all participant providing consent for genetic analyses, N = 20/34 were allocated to the high-dose vitamin $D_3$ supplementation arm (Fig 1). Of these 20 participants, three had no biomaterial available for 25(OH)D analysis by the LC-MS/MS method. For these three participants, 25(OH)D levels were available as determined with the immunoassay as used in the SOLAR trial [13]. Since these 3 individuals had week 48 levels of 25(OH)D below the assay

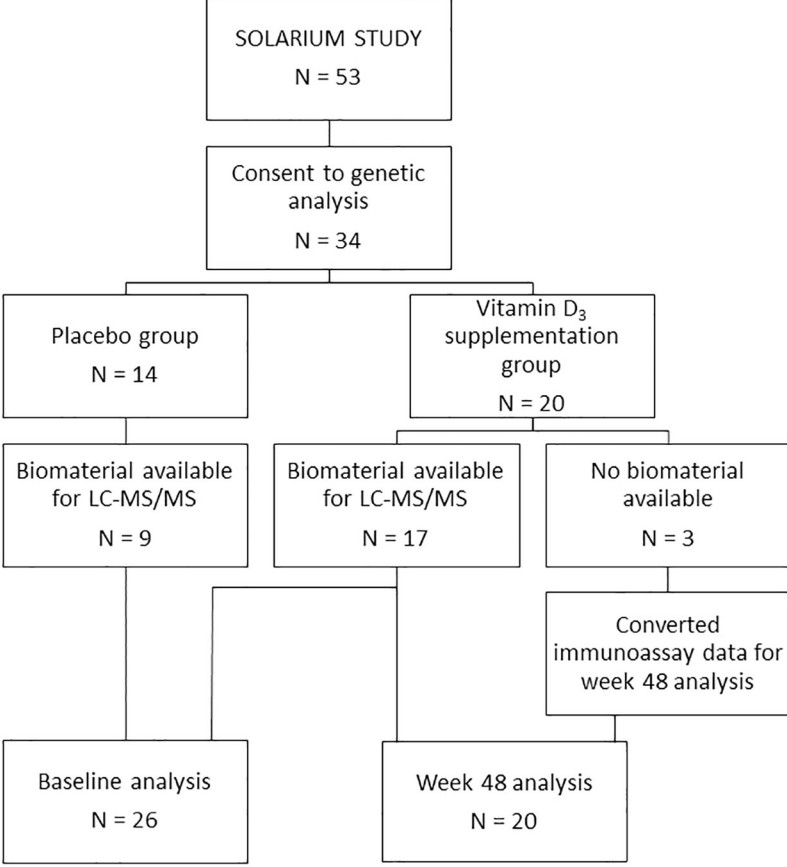

**Fig 1. Flowchart depicting availability of data for our analyses.** LC-MS/MS: Liquid chromatography-tandem mass spectrometry.

upper detection limit (<250 nmol/L), and we earlier reported a good correlation between the immunoassay and LC-MS/MS within this range (R = 0.917, [21]), we converted for these individuals their 25(OH)D status using a linear model.

## 2.3 Defining SNPs of interest

We selected for our study two SNPs that are missense variants in the gene encoding vitamin D-binding protein (DBP, or Group Component) (rs4588 and rs7041), as well as one SNP near the CYP27B1 gene (rs12368653) and one SNP near the CYP24A1 gene (rs2248359), which are all related to vitamin D metabolism. SNPs rs12368653 and rs2248359 have been identified as MS risk-alleles in a recent large GWAS [7]. The DBP-associated SNPs rs4588 and rs7041 have been reported as top-hits in several GWAS studies on determinants of 25(OH)D status in healthy and diseased cohorts [22–24]. Although not all GWAS studies support these specific DBP SNPs, DBP has been identified in all GWAS as a locus influencing 25(OH)D levels [25, 26]. Since the rs7041 and rs4588 SNPs were also associated with response to low-dose vitamin D supplementation in non-MS cohorts [27–29], we included these SNPs in our analysis. We genotyped rs2248359, the others were imputed using the RICOPILI pipeline and the Michigan server (HRC-reference panel). Genotyping was performed using extracted DNA from buffy coats. This DNA was analysed on the Infinium PsychArray-24v1.3_A1 BeadChip (Infinium array technology) and the polymorphisms of interest were defined through standard protocols.

## 2.4 Statistical analysis

SPSS software (IBM SPSS, version 25.0. Chicago, IL) was used to evaluate the differences in vitamin D status between carriers of risk alleles versus non-carriers. Normality of data was assessed by visual inspection of normality plots, as well as the Shapiro-Wilk test. To compare baseline characteristics, Mann-Whitney U tests were used to compare continuous data, while chi square tests were used to compare dichotomous data. Because of sample sizes, a Mann-Whitney U test was used to compare 25(OH)D levels between carriers and non-carriers. Previously, we have shown BMI to be a confounder in vitamin D levels in MS patients. Therefore, we have marked all patients with a BMI $\geq 25 \text{kg/m}^2$ in our analyses. A p-value of <0.05 was considered statistically significant.

# 3. Results

## 3.1 Baseline characteristics

In the original SOLARIUM trial, 53 participants completed the 48 week follow-up. Of these participants, 34 participants consented to genetic analysis; 14 participants received placebo and 20 participants received high-dose vitamin $D_3$ supplementation. Regarding vitamin D assays, 26 of the 34 participants had material available for the LC-MS/MS method at baseline and at week 48. Of these 26 participants, 9 received placebo and 17 received high-dose vitamin $D_3$ supplementation. For the 3 remaining participants that received high-dose vitamin $D_3$ supplementation, immunoassay data were available and these were converted as described in section 2.2 (Fig 2). After a 48 week follow-up, 6 of the 26 participants showed signs of MRI activity, 19 showed no signs of MRI activity and 1 was unavailable for MRI analysis. For all baseline characteristics, as well as treatment arm and MRI outcome, no statistically significant difference was observed between the SOLARIUM group and this subcohort (Table 1).

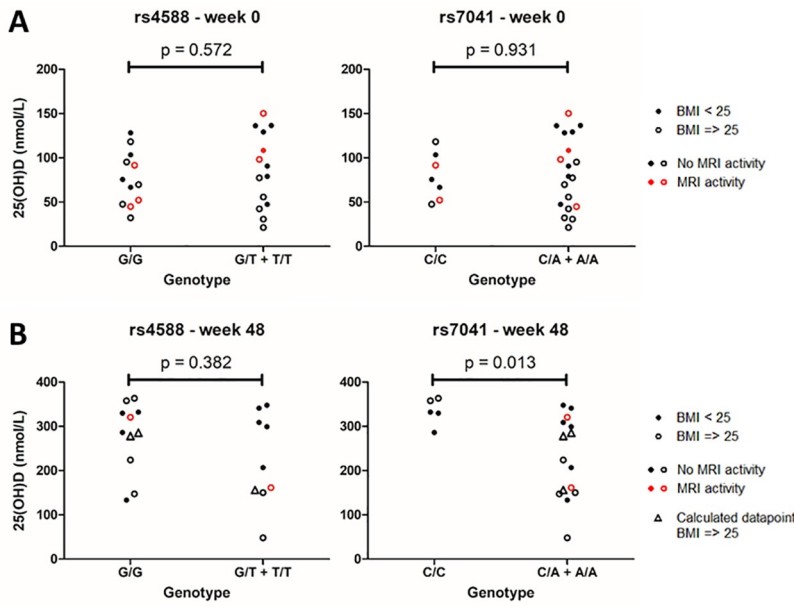

**Fig 2. Differences in 25(OH)D levels between carriers and non-carriers of DBP related risk alleles.** 25(OH)D levels at baseline (N = 26) (**A**) and after 48 weeks of supplementation (N = 20) (**B**), compared between non-carriers (left group) and carriers (right group) of the rs4588 and rs7041 risk alleles. P-value shown is calculated using a Mann-Whitney U test. Triangles indicate calculated data, derived from immunoassay values to replace missing values from the LC-MS/MS method.

**Table 1. Baseline characteristics of the studied participants.**

|  | Participants with available genetic material (N = 26*) | Total SOLARIUM population (N = 53*) | p-value |
|---|---|---|---|
| **Sex (N[%])** |  |  | 0.777 |
| Female | 18 [69] | 35 [66] |  |
| Male | 8 [31] | 18 [34] |  |
| **Age (years: median [interquartile range]** | 39.9 [32.6–45.1] | 36.2 [31.4–43.9] | 0.591 |
| **Body Mass Index (BMI) (N[%])** |  |  | 0.802 |
| $< 25 \ kg/m^2$ | 11 [42] | 24 [45] |  |
| $\geq 25 \ kg/m^2$ | 15 [58] | 29 [55] |  |
| **Disease duration (months: median [interquartile range])** | 7.3 [5.2–11.7] | 7.3 [4.4–11.8] | 0.900 |
| **Attacks during past 2 years at baseline (N[%])** |  |  | 0.691 |
| $\leq 1$ | 17 [65] | 37 [70] |  |
| $> 1$ | 9 [35] | 16 [30] |  |
| **Time since last attack at baseline (months: median [interquartile range])** | 7.4 [4.6–11.3] | 7.5 [5.0–10.4] | 0.983 |
| **Treatment (N[%])** |  |  | 0.455 |
| Placebo | 9 [35] | 23 [43] |  |
| Vitamin D3 | 17 [65] | 30 [57] |  |
| **MRI activity after 48 weeks follow-up (N[%])** |  |  | 0.851 |
| No MRI activity | 19 [76] | 37 [74] |  |
| MRI activity | 6 [24] | 13 [26] |  |

P-value is based on Mann-Whitney U test for continuous data and Chi square test for dichotomous data.

* MRI data after 48 weeks follow-up were available for N = 25 participants from the genetic material group and N = 50 participants from the total SOLARIUM group.

## 3.2 Vitamin D binding protein SNP rs7041 shows increased serological response to supplementation

DBP is a known important determinant of 25(OH)D status [30]. Two missense SNPs in the DBP gene, rs4588 and rs7041, were imputed and 25(OH)D levels were compared between carriers and non-carriers of the allele associated with lower 25(OH)D levels. Baseline 25(OH)D levels did not significantly differ between the two groups (Fig 2A). However, after 48 weeks of high-dose vitamin $D_3$ supplementation, carriers of the rs7041 risk allele showed a lower serological response compared to non-carriers [median(IQR): 224.2 nmol/L (150.1–308.9) and 332.0 nmol/L (329.9–357.7), respectively, p = 0.013] (Fig 2B). Accordingly, carriers of the rs7041 risk allele showed a reduced absolute increase of 25(OH)D compared to non-carriers [median(IQR): 148.9 nmol/L (44.6–208.2) and 245.3 nmol/L (226.5–265.3), respectively, p = 0.020].

## 3.3 CYP27B1-related SNP rs12368653 associated with higher vitamin D level after supplementation

Then, two known MS risk alleles that are related to vitamin D metabolism, rs12368653 (near *CYP27B1*) and rs2248359 (near *CYP24A1*) [7], were analysed for differences in 25(OH)D levels. Again, baseline 25(OH)D values did not significantly differ between carriers and non-carriers of risk alleles (Fig 3A). After 48 weeks of supplementation, carriers of the rs12368653 risk allele showed higher levels of 25(OH)D compared to non-carriers [median(IQR): 304.1 nmol/L (251.2–336.7) and 152.0 nmol/L (140.6–158.9), respectively, p = 0.014] (Fig 3B). Additionally, carriers showed a higher absolute increase in 25(OH)D levels over 48 weeks compared to non-carriers [median(IQR): 211.0 nmol/L (170.0–261.5) and 11.2 nmol/L (5.5–77.9),

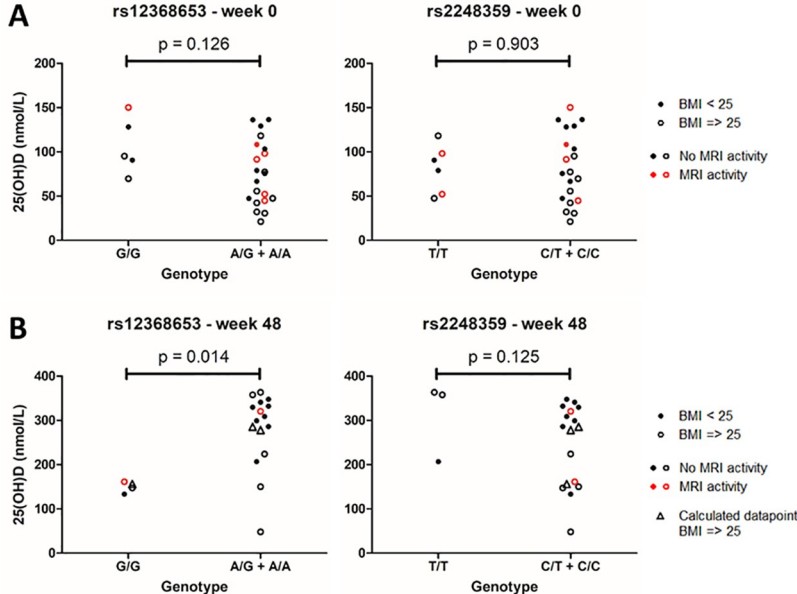

**Fig 3. Differences in 25(OH)D levels between carriers and non-carriers of CYP related MS risk alleles.** 25(OH)D levels at baseline (N = 26) (**A**) and after 48 weeks vitamin $D_3$ supplementation (N = 20) (**B**), compared between non-carriers (left group) and carriers (right group) of the rs12368653 and rs2248359 risk alleles. P-value shown is calculated using a Mann-Whitney U test. Triangles indicate calculated data, derived from immunoassay values to replace missing values from the LC-MS/MS method.

respectively, p = 0.023]. 25(OH)D levels did not significantly differ between carriers and non-carriers of the rs2248359 risk allele.

## 4. Discussion

We explored the effect of vitamin D-related genetics on serological response to high-dose vitamin $D_3$ supplementation in a cohort of relapsing-remitting MS patients treated with inter-feron-β-1a. We report an association between two genetic markers and serological response to high-dose vitamin $D_3$ supplementation, where the *DBP*-linked rs7041 risk allele shows a decreased serological response, and the *CYP27B1*-linked rs12368653 risk allele shows an increased response in circulating 25(OH)D levels.

Studies investigating the DBP rs7041 risk allele and 25(OH)D status show varying results. A study by Sollid et al. [30] showed a difference in 25(OH)D at baseline, but not after supplementation. By contrast, a recent study by Al-Daghri et al. [27] showed no difference in 25(OH)D levels at baseline between rs7041 genotypes, but did show a decreased serological response to 6 months of vitamin $D_3$ supplementation for carriers of the risk allele. Our findings add to the latter observation, showing a relevant effect of genetic background on serological response to vitamin D supplements.

CYP27B1 (1α-hydroxylase) catalyses the hydroxylation of the inactive 25(OH)D to the active 1,25(OH)$_2$D. The identification of the rs12368653 risk-SNP for MS further supports the role of vitamin D in developing MS. Our findings showed an increased serological response to vitamin $D_3$ supplementation in carriers of the risk allele. We speculate that this finding may reflect a more strict regulation of vitamin D metabolism in carriers of de *CYP27B1*-linked risk-allele. At present, no experimental data addressing the functional effects of this SNP on 1-α hydroxylation of vitamin D have been published. These findings appear contradictory with the findings from Bhargava et al. that MS is related to a decreased response to supplementation. One explanation for this may lie in the fact that the rs12368653 risk allele shows an odds ratio for developing MS of 1.1 [7]. As such, the contribution of rs12368653 to the development of MS is very limited. Additionally, our findings are found in supra-physiological conditions and thus differ from the conditions in which MS develops. Nevertheless, differing alleles in rs12368653 appear to influence vitamin D metabolism in a biologically relevant manner, reducing the serological response to supplementation in the SOLARIUM study. As such, rs12368653 may be a relevant confounder in supplementation studies.

In our study, we find no difference in 25(OH)D levels before and after supplementation between carriers and non-carriers of both the rs4588 and rs2248359 risk alleles. However, both have been associated with 25(OH)D levels in other studies, with the rs4588 risk allele showing a decreased serological response to vitamin $D_3$ supplementation [27], and the rs2248359 risk allele showing a decreased baseline level of 25(OH)D [31]. This discrepancy may be explained by our relatively small sample size, which increases the risk of type 2 errors.

Our findings suggest that genetic components have influenced the serological response to vitamin $D_3$ supplementation in the SOLARIUM study and may be a relevant confounder in general supplementation studies. However, it is currently unknown how this translates to clinical outcome in MS patients. As lower 25(OH)D levels have been linked to an increased risk of disease activity [6], a relation between the serological effect of genetics and clinical outcome could be hypothesised. Additionally, findings by Graves et al. [16] show that children with a unfavourable genetic composition regarding vitamin D metabolism show an increased risk for relapses, which points towards a relevant relation between vitamin D genetics and clinical outcome in MS. Alternatively, genetic background could also influence the occurrence of adverse events of high-dose vitamin $D_3$ supplementation, such as hypercalcaemia. Although

hypercalcaemia could negatively affect the disease course of MS [32], hypercalcaemia has not been observed in high-dose vitamin $D_3$ supplementation studies thus far [33]. Taken together, the influence of genetics on the clinical response to vitamin $D_3$ supplementation should be investigated further. It is also important to mention that our findings show an effect on supra-physiological levels of 25(OH)D after high-dose vitamin $D_3$ supplementation. It remains to be determined whether the reported influences of rs7041 and rs12368653 remain relevant with more physiological levels of vitamin $D_3$ supplementation.

Strengths of our study include its double blind nature [18]. This study is limited by its relatively small sample size, which also made it impossible to analyse each SNP separately, and instead having to use a dichotomous carrier/non-carrier designation for risk alleles.

In conclusion, the vitamin D-related genetic background influences the serological response to high-dose vitamin $D_3$ supplementation and, as such, may need to be corrected for in later supplementation studies. The clinical consequence of this altered serological response should be investigated further.

## Supporting information

**S1 Table. Baseline characteristics, genetic composition and vitamin D levels of all SOLAR-IUM participants.**
(XLSX)

## Acknowledgments

We would like to thank Ward de Witte for his expertise and assistance in genotyping and imputation.

## Author Contributions

**Conceptualization:** Raymond Hupperts, Jan Damoiseaux, Joost Smolders.

**Data curation:** Max Mimpen, Linda Rolf, Geert Poelmans, Jody van den Ouweland, Raymond Hupperts.

**Formal analysis:** Max Mimpen, Jan Damoiseaux, Joost Smolders.

**Funding acquisition:** Raymond Hupperts, Joost Smolders.

**Investigation:** Max Mimpen, Jan Damoiseaux.

**Methodology:** Max Mimpen, Linda Rolf, Geert Poelmans, Jody van den Ouweland, Raymond Hupperts.

**Project administration:** Raymond Hupperts, Jan Damoiseaux, Joost Smolders.

**Resources:** Linda Rolf, Geert Poelmans, Jody van den Ouweland, Raymond Hupperts.

**Supervision:** Raymond Hupperts, Jan Damoiseaux, Joost Smolders.

**Validation:** Jody van den Ouweland.

**Visualization:** Max Mimpen.

**Writing – original draft:** Max Mimpen, Jan Damoiseaux, Joost Smolders.

**Writing – review & editing:** Linda Rolf, Geert Poelmans, Jody van den Ouweland, Raymond Hupperts.

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
