## [Decision Letter · Decision Letter 0]

19 Oct 2021

PONE-D-21-25514Vitamin D related genetic polymorphisms affect serological response to high-dose vitamin D supplementation in multiple sclerosisPLOS ONE

Dear Dr. Smolders,

Thank you for submitting your manuscript to PLOS ONE. After careful consideration, we feel that it has merit but does not fully meet PLOS ONE’s publication criteria as it currently stands. Therefore, we invite you to submit a revised version of the manuscript that addresses the points raised during the review process.

Specifically, caution regarding clinical relevance of these findings was suggested given the small samples size of the study.

We look forward to receiving your revised manuscript.

Kind regards,

Tobias Derfuss

Academic Editor

PLOS ONE

Journal Requirements:

“MM has nothing to disclose; LR has nothing to disclose; GP is director of Drug Target ID, Ltd.; JO has nothing to disclose; RH received institutional research grants and fees for lectures and advisory boards from Biogen, Merck, and Genzyme-Sanofi; JD has nothing to disclose; JS received lecture and/or consultancy fees from Biogen, Merck, Sanofi-Genzyme, and Novartis.”

Reviewers' comments:

Reviewer's Responses to Questions

**Comments to the Author**

1. Is the manuscript technically sound, and do the data support the conclusions?

Reviewer #1: Yes

Reviewer #2: Yes

2. Has the statistical analysis been performed appropriately and rigorously? 

Reviewer #1: Yes

Reviewer #2: Yes

3. Have the authors made all data underlying the findings in their manuscript fully available?

Reviewer #1: Yes

Reviewer #2: Yes

4. Is the manuscript presented in an intelligible fashion and written in standard English?

Reviewer #1: Yes

Reviewer #2: Yes

5. Review Comments to the Author

Reviewer #1: This manuscripts contains a secondary analysis from the SOLAR RCT and its substudy SOLARIUM. It explores whether vitamin D-associated SNPs and MS risk alleles affect serological response to vitamin D supplementation. The authors report that some vitamin D-related SNPs modulate the serological response to high-dose vitamin D supplementation, which might explain the partly contradictory results of previous vitamin D supplementation studies.

Strengths of the study are the relevant research question, adequate methodology and the interesting results, which give a potential mechanistic explanation for differential clinical effects of vitamin D supplementation.

Limitations are the small sample size of just 26 cases of the original 229 patients in the SOLAR study and 53 patients of the SOLARIUM substudy. Thus, it is questionable if the current study can be used for interpretation of the (negative) clinical results of the vitamin D supplementation study, as claimed by the authors in the introduction.

Further points to consider:

1. The original study was funded by Merck. Did the sponsor also have a role in this subanalysis?

2. Were all currently known polymorphisms related to the vitamin D status in MS examined?

3. Were the observed differences in 25(OH)D levels related to specific SNPs associated with differences in immune cell frequencies or function (e.g. Treg, Teff) or cytokine levels in the serum?

Reviewer #2: This is an interesting manuscript investigating whether the serum response to Vitamin D supplementation is affected by the respective genetic background. The associations are sound although far from being conclusive or causal. The discussion should be widenend a bit towards the emerging concept that high dose Vitamin D may also raise serum calcium, which may not be of benefit (Häusler, BRAIN, 2019).

6. PLOS authors have the option to publish the peer review history of their article (what does this mean?). If published, this will include your full peer review and any attached files.

Reviewer #1: No

Reviewer #2: No

---

## [Author Response · Author response to Decision Letter 0]

4 Nov 2021

We thank the reviewers for their feedback. Below is a list of the reviewers’ comments, together with our responses. At the end of each response we listed the line where the alteration is located in the tracked changes document.

Reviewer #1

The original study was funded by Merck. Did the sponsor also have a role in this subanalysis?

A: Merck was not involved in the gathering, analysis or interpretation of these results. We have added an additional statement in the acknowledgments section to specify this. Line 185-187

“This study was funded by Nationaal MS Fonds grant OZ2016-001 and an unrestricted grant by Merck. Neither sponsor played any role in the gathering, analysis or interpretation of the presented data.”

Were all currently known polymorphisms related to the vitamin D status in MS examined?

A: Acknowledging the small sample-size of our dataset, we anticipated to restrict our study to 1) the most reproduced 25(OH)D status-associated SNPs in the general population, and 2) the vitamin D-associated MS risk alleles of the IMSGC Nature 2011 paper. The reviewer makes a valid point that we should put more emphasis on the selection of the SNPs for the current project in our methods section. Therefore, we added the argumentation above including references to the manuscript. Paragraph 2.3.

“For our study, we selected two SNPs that are missense variants in the gene encoding vitamin D-binding protein (DBP, or Group Component) (rs4588 and rs7041), as well as one SNP near the CYP27B1 gene (rs12368653) and one SNP near the CYP24A1 gene (rs2248359), which are all related to vitamin D metabolism. SNPs rs12368653 and rs2248359 have been identified as MS risk-alleles in a recent large GWAS.[1] The DBP-associated SNPs rs4588 and rs7041 have been reported as top-hits in several GWAS studies on determinants of 25(OH)D status in healthy and diseased cohorts.[2-4] Although not all GWAS studies support these specific DPB SNPs, DBP has been identified in all GWAS as a locus influencing 25(OH)D levels.[5, 6] Since the rs7041 and rs4588 SNPs were also associated with response to low-dose vitamin D supplementation in non-MS cohorts,[7-9] we included these SNPs in our analysis. Genotyping was performed using extracted DNA from buffy coats. This DNA was analysed on the Infinium PsychArray-24v1.3_A1 BeadChip (Infinium array technology) and the polymorphisms of interest were defined through standard protocols. We genotyped rs2248359, the others were imputed using the RICOPILI pipeline and the Michigan server (HRC-reference panel).”

Were the observed differences in 25(OH)D levels related to specific SNPs associated with differences in immune cell frequencies or function (e.g. Treg, Teff) or cytokine levels in the serum?

A: As mentioned in the original SOLARIUM study, 25(OH)D levels did not significantly affect T cell frequencies, function or cytokine levels. Similarly, recent publications of our group revealed no significant effect of 25(OH)D supplementation on NK cell or B cell frequencies. Although this analysis falls outside the scope of the present study, adding the mentioned vitamin D related SNPs as confounding factors did not alter these previous results.

Reviewer #2

This is an interesting manuscript investigating whether the serum response to Vitamin D supplementation is affected by the respective genetic background. The associations are sound although far from being conclusive or causal. The discussion should be widened a bit towards the emerging concept that high dose Vitamin D may also raise serum calcium, which may not be of benefit (Häusler, BRAIN, 2019).

A: We thank the reviewer for this suggestion. Our results do indeed allow for a wider discussion about the topic of high-dose vitamin D3 supplementation, including its possible negative effects. We have added a paragraph to address this topic. Line: 171-175

“ Alternatively, genetic background could also influence the occurrence of adverse events of high-dose vitamin D3 supplementation, such as hypercalcaemia. Although hypercalcaemia could negatively affect the disease course of MS,[10] hypercalcaemia has not been observed in high-dose vitamin D3 supplementation studies thus far.[11]”

1. International Multiple Sclerosis Genetics, C., et al., Genetic risk and a primary role for cell-mediated immune mechanisms in multiple sclerosis. Nature, 2011. 476(7359): p. 214-9.

2. Ahn, J., et al., Genome-wide association study of circulating vitamin D levels. Hum Mol Genet, 2010. 19(13): p. 2739-45.

3. Engelman, C.D., et al., Genetic and environmental determinants of 25-hydroxyvitamin D and 1,25-dihydroxyvitamin D levels in Hispanic and African Americans. J Clin Endocrinol Metab, 2008. 93(9): p. 3381-8.

4. O'Brien, K.M., et al., Genome-Wide Association Study of Serum 25-Hydroxyvitamin D in US Women. Front Genet, 2018. 9: p. 67.

5. Jiang, X., et al., Genome-wide association study in 79,366 European-ancestry individuals informs the genetic architecture of 25-hydroxyvitamin D levels. Nat Commun, 2018. 9(1): p. 260.

6. Wang, T.J., et al., Common genetic determinants of vitamin D insufficiency: a genome-wide association study. Lancet, 2010. 376(9736): p. 180-8.

7. Al-Daghri, N.M., et al., Efficacy of vitamin D supplementation according to vitamin D-binding protein polymorphisms. Nutrition, 2019. 63-64: p. 148-154.

8. Enlund-Cerullo, M., et al., Genetic Variation of the Vitamin D Binding Protein Affects Vitamin D Status and Response to Supplementation in Infants. J Clin Endocrinol Metab, 2019. 104(11): p. 5483-5498.

9. Ganz, A.B., et al., Vitamin D binding protein rs7041 genotype alters vitamin D metabolism in pregnant women. FASEB J, 2018. 32(4): p. 2012-2020.

10. Hausler, D., et al., High dose vitamin D exacerbates central nervous system autoimmunity by raising T-cell excitatory calcium. Brain, 2019. 142(9): p. 2737-2755.

11. Smolders, J., J. Damoiseaux, and R. Hupperts, Hypercalcaemia rather than high dose vitamin D3 supplements could exacerbate multiple sclerosis. Brain, 2019. 142(12): p. e71.

---

## [Editor Report · Decision Letter 1]

24 Nov 2021

Vitamin D related genetic polymorphisms affect serological response to high-dose vitamin D supplementation in multiple sclerosis

PONE-D-21-25514R1

Dear Dr. Smolders,

We’re pleased to inform you that your manuscript has been judged scientifically suitable for publication and will be formally accepted for publication once it meets all outstanding technical requirements.

Kind regards,

Tobias Derfuss

Academic Editor

PLOS ONE

---

## [Editor Report · Acceptance letter]

25 Nov 2021

PONE-D-21-25514R1 

Vitamin D related genetic polymorphisms affect serological response to high-dose vitamin D supplementation in multiple sclerosis 

Dear Dr. Smolders:

I'm pleased to inform you that your manuscript has been deemed suitable for publication in PLOS ONE. Congratulations! Your manuscript is now with our production department. 

Kind regards, 

on behalf of

Dr. Tobias Derfuss 

Academic Editor

PLOS ONE